# The Ascyrus Medical Dissection Stent: A One-Fits-All Strategy for the Treatment of Acute Type A Aortic Dissection?

**DOI:** 10.3390/jcm13092593

**Published:** 2024-04-28

**Authors:** Leonard Pitts, Michael C. Moon, Maximilian Luehr, Markus Kofler, Matteo Montagner, Simon Sündermann, Semih Buz, Christoph Starck, Volkmar Falk, Jörg Kempfert

**Affiliations:** 1Deutsches Herzzentrum der Charité (DHZC), Department of Cardiothoracic and Vascular Surgery, Augustenburger Platz 1, 13353 Berlin, Germany; markus.kofler@dhzc-charite.de (M.K.); matteo.montagner@dhzc-charite.de (M.M.); simon.suendermann@dhzc-charite.de (S.S.); semih.buz@dhzc-charite.de (S.B.); christoph.starck@dhzc-charite.de (C.S.); volkmar.falk@dhzc-charite.de (V.F.); joerg.kempfert@dhzc-charite.de (J.K.); 2Charité—Universitätsmedizin Berlin, Corporate Member of Freie Universität Berlin and Humboldt-Universität zu Berlin, Charitéplatz 1, 10117 Berlin, Germany; 3Division of Cardiac Surgery, University of Alberta, Edmonton, AB T6G 1H9, Canada; mmoon@ualberta.ca; 4Department of Cardiothoracic Surgery, Heart Centre, University of Cologne, 50923 Cologne, Germany; maximilian.luehr@uk-koeln.de; 5DZHK (German Centre for Cardiovascular Research), Partner Site Berlin, 10785 Berlin, Germany; 6Translational Cardiovascular Technologies, Institute of Translational Medicine, Department of Health Sciences and Technology, Swiss Federal Institute of Technology (ETH), 8092 Zurich, Switzerland

**Keywords:** acute type A aortic dissection, aorta, Ascyrus Medical Dissection Stent, endovascular, malperfusion, frozen elephant trunk, thoracic endovascular aortic repair

## Abstract

The treatment of DeBakey type I aortic dissection remains a major challenge in the field of aortic surgery. To upgrade the standard of care hemiarch replacement, a novel device called an “Ascyrus Medical Dissection Stent” (AMDS) is now available. This hybrid device composed of a proximal polytetrafluoroethylene cuff and a distal non-covered nitinol stent is inserted into the aortic arch and the descending thoracic aorta during hypothermic circulatory arrest in addition to hemiarch replacement. Due to its specific design, it may result in a reduced risk for distal anastomotic new entries, the effective restoration of branch vessel malperfusion and positive aortic remodeling. In this narrative review, we provide an overview about the indications and the technical use of the AMDS. Additionally, we summarize the current available literature and discuss potential pitfalls in the application of the AMDS regarding device failure and aortic re-intervention.

## 1. Introduction

Acute type A aortic dissection (ATAAD) is associated with poor outcomes [1]. Although there has been great progress in terms of surgical and perioperative technologies over the last decades, morbidity and mortality are still high [2]. Advanced age and preoperative malperfusion contribute significantly to a higher surgical risk, considering that open surgery remains the therapy of choice in the treatment of ATAAD [3,4]. The resection of the entry tear in combination with an open distal anastomosis under adequate cerebral protection is recommended to prevent aortic rupture, reestablish antegrade flow in the true lumen and resolve malperfusion [5,6]. According to the current guidelines, this includes the resection of the ascending aorta at least in terms of a hemiarch replacement [7,8]. Though this strategy effectively treats the ascending aorta, the aortic arch and descending aorta remain untouched in the case of DeBakey type I dissection. Especially in the case of supra-aortic vessel involvement or consecutive aortic branch vessel malperfusion, more extensive repair may be advantageous [9]. For this scenario, the Ascyrus Medical Dissection Stent (AMDS; Artivion^®^, Atlanta, GA, USA) was developed to upgrade the standard of care hemiarch procedure, aiming to reduce complications deriving from true lumen collapse and false lumen patency [10,11].

## 2. Device Description and Surgical Procedure

The AMDS is a hybrid prosthesis consisting of a proximal cuff composed of polytetrafluorethylene and an uncovered superhelical nitinol stent (Figure 1).

The aim of the cuff is to effectively seal the distal anastomosis and prevent false lumen flow as well as to lower the risk for distal anastomotic new entries (DANEs) [12]. Currently, two cuff sizes are available: 24 and 32 mm. The stent frame was designed for true lumen stabilization and consecutive positive aortic remodeling in the aortic arch and the downstream aorta. Flexibility is allowed to adapt to the curvature of the aortic arch, enabling insertion in zone 0. Two different shapes of the nitinol stent are currently available (straight and tapered) and four different sizes. The choice for the adequate size is made through an evaluation of a preoperative computed tomography scan and pragmatic measurement of the diameter at two aortic landmarks: zone 1 (aortic arch) and zone 4 (tracheal bifurcation). The appropriate stent can then be selected according to the sizing charts provided by the manufacturer. The device is simple to handle and does not significantly prolong the surgical procedure [13]. A corresponding video demonstrating the step-by-step implantation and providing surgical tips and tricks was recently published by our group [10]. A schematic of the procedure for implantation is shown in Figure 2. 

## 3. When to Use AMDS—And When to Avoid It

As already mentioned, the AMDS was conducted to upgrade the standard of care hemiarch procedure. Rylski et al. showed that the incidence for DANEs after hemiarch replacement in the setting of ATAAD may occur in up to 70% of cases [14]. False lumen perfusion (*p* < 0.001) and DANEs (*p* < 0.001) were strongly associated with the increased growth of the residual dissected aorta, which is a well-studied risk factor for further aortic-related interventions and death [14,15]. This is consistent with other studies, showing that a patent false lumen caused by DANEs after ATAAD repair shows greater aortic growth rate of the descending aorta and is one of the leading risk factors for distal aortic events [16]. The first try to address false lumen patency was the so-called Djumbodis Dissection System, a non-self-expanding bare metal stent, which was deployed into the aortic arch in addition to hemiarch replacement [17]. However, most of the patients had continuing antegrade perfusion of the false lumen, and the authors concluded that under these circumstances, no additive value as an adjunct to hemiarch replacement exists, most likely due to the non-self-expanding capability of the device. The main differences between the Djumbodis Dissection System and AMDS may be the non-availability of the proximal cuff to avoid the formation of DANEs and stent migration, as well as the non-self-expanding stent capability, which favors false lumen patency. Besides the low numbers of implantations, available data on the long-term outcome of patients who received an additional implantation of the Djumbodis Dissection Stent are limited to a minimum. Vendramin et al. summarized the long-term follow up and late complications of patients treated with the Djumbodis Dissection Stent and discovered high rates of late complications associated with the device, among them were stent fracture and stent migration [18]. The authors not only determined the insufficiency of this device, but also advised that patients with a Djumbodis Dissection Stent undergo regular monitoring to mitigate the risk of potential catastrophic incidents stemming from device failure.

This highlights the urgent need for an appropriate tool to address the challenges of the residual dissected aorta. However, there are a few key points (Table 1) that should be considered to identify potential candidates who benefit from additional AMDS implantations. 

There is an ongoing debate on whether to perform hybrid arch repair using a frozen elephant trunk (FET) or the AMDS [19]. It must be stated clearly that these prostheses are two different kinds of animals, and therefore, their indications are also different. Indeed, the FET represents an excellent treatment option in case of DeBakey type I dissection with consecutive malperfusion. Outstanding results have been published in the past, bearing in mind that these data were derived from specialized aortic centers with corresponding expertise in the use of FET for total arch replacement [20,21]. Though representing the gold standard for definite and complete arch repair, it requires a professional aortic team with experienced aortic surgeons to achieve satisfactory results and low perioperative mortality rates because of its high complexity [22]. Performing a FET procedure in the setting of ATAAD may not be feasible for every surgeon on-call without advanced aortic surgery training and experience. Especially for this scenario, additional AMDS implantations and hemiarch replacements represent valid alternatives in case of a life-saving operation for DeBakey type I dissection. However, if contraindications exist, no compromises should be made, and total arch replacement, preferably using a FET, should be performed [23]. According to our expertise, entries not only in the aortic arch or descending aorta, but also in the supra-aortic vessels may contribute to a perfused false lumen after AMDS implantation, leading to aortic growth and a high risk for complex redo surgery. This highlights the importance of preoperative planning including multiplanar computed tomography reconstructions to assess the individual dissection patterns precisely and offer adequate aortic repair [1]. In a nutshell, the AMDS does not replace the FET, but offers a valid alternative in the case of DeBakey type I dissection and the absence of specific contraindications while upgrading the hemiarch procedure. In the case of chronic aortic dissection, no evidence is currently available in terms of AMDS implantation, and therefore, no reliable recommendations can be provided so far. According to our experience, AMDS implantation should not be considered for the treatment of chronic aortic dissection, e.g., of the aortic arch.

## 4. Current Clinical Results

The first results following AMDS implantation for the treatment of ATAAD were published by Boszo et al. in the “Dissected Aorta Repair Through Stent Implantation” (DARTS) trial [24]. In this safety and feasibility multicenter study, 16 patients with DeBakey type I dissection were enrolled, of whom 50% had evidence of preoperative malperfusion. The thirty-day mortality was 6.3%, and complete or partial thrombosis, including the remodeling of the aortic arch and descending aorta, was detected in 91.7% of cases (*n* = 11/12 with complete follow-up). Though the median follow-up time was only 130 ± 94 days, the results were promising and paved the way for further investigation. The DARTS trial expanded, and more centers participated, enrolling a total of 47 patients between 2017 and 2019 with a median follow-up time of 631 days [15]. Preoperative malperfusion was present in 56.5% of patients, including three cases of spinal malperfusion with consecutive paraplegia. The thirty-day mortality was 13%, and there were no device-related complications. The complete obliteration or thrombosis of the false lumen was observed in 74% in the aortic arch and in 53% in the descending aorta. Spinal malperfusion resolved in all cases. The AMDS promoted false lumen closure at the distal anastomosis in 90% of patients. A further sub-analysis of this cohort revealed excellent results for the restoration of malperfusion: 95.5% (n = 63) of branch vessel malperfusion cases resolved without an additional procedure [25]. New perioperative stroke, defined by the absence of preoperative cerebral malperfusion, occurred in 7.7% (n = 2) of patients. Preoperative cerebral malperfusion caused by the dissection of supra-aortic vessels is especially crucial and significantly increases the risk of perioperative stroke [5]. Current evidence is limited, but in a series of 16 patients, we were able to demonstrate satisfactory results in terms of supra-aortic vessel restoration and reached 100% regression of the totally occluded supra-aortic branches after the AMDS was implanted [26]. Later on, we published the currently largest available series of AMDS implantations for the treatment of DeBakey type I dissection, which includes 100 patients [6]. The thirty-day mortality and the rate of new postoperative stroke were 18% and 8%, respectively. Technical success was achieved in 76%, defined as the induced thrombosis of the false lumen in the medial segment of the descending aorta. Unfortunately, results for the long-term follow-up of aortic remodeling and false lumen patency are still missing. Recently, the three-year outcomes of the DARTS trial were published: the false lumen was completely or partially thrombosed in 90.5% in zone 0, 60.0% in zone 1, 68.2% in zone 2 and 89% in zone 5 [27]. Though AMDS was designed for zone 0 insertion, the partial replacement of the aortic arch and more distal implantation may be necessary in selected scenarios. Mehdiani et al. investigated the outcomes of eight patients receiving AMDS implantation beyond zone 0 [28]. No malperfusion was present in the survivors (7/8 patients), and true lumen was open in all patients, while the true lumen area was significantly higher in zone III (*p* = 0.016) and at the level of T11 (*p* = 0.009). The authors concluded that additional AMDS implantation beyond zone 0 can be safely performed and that it potentially avoids the risk for spinal cord injury, which is a rare but serious complication in the case of FETs [29]. Another series with 57 patients was published by Luehr et al., demonstrating an in-hospital mortality of 16% and a new postoperative stroke rate of 4%, which are in line with previous results [30]. Justified criticism has been raised about the additional use of an AMDS, questioning its potential benefit against standard hemiarch replacement, considering that the AMDS is way more expensive than a single dacron graft [31]. The first study comparing outcomes between single hemiarch replacement and additional AMDS implantation was recently published, investigating the impacts on aortic remodeling and risk for DANEs [12]. In this retrospective dual-center trial, 114 patients met the inclusion criteria and underwent hemiarch replacement in case of DeBakey type I dissection, whereas 37 patients received additional AMDS implantation. Despite no difference in mortality (*p* = 0.768) or other in-hospital adverse events, the incidence for DANEs was significantly lower with 11.8% (n = 4) in the AMDS group compared to 43.3% (n = 26) in the isolated hemiarch group (*p* = 0.002). Additionally, positive aortic remodeling in terms of false lumen thrombosis was superior in the AMDS group at the level of the aortic arch (*p* = 0.029), the proximal descending aorta (*p* = 0.031) and the level of pulmonary artery bifurcation (*p* = 0.044). These preliminary results are promising and suggest a broader application of the AMDS, considering that long-term follow-up data are still missing. The latest results of the DARTS trial and currently available studies (except case reports or case series) investigating outcomes after AMDS implantation are summarized in Table 2.

## 5. Potential Risk for Device Failure

Due to the small number of implantations, current experience about potential device failure is limited. In their series of 57 AMDS implantations, Luehr et al. discovered that in 5 patients (8%), the proximal and distal AMDS portions were inflated while a complete central stent collapse was evident [30]. An example of AMDS collapse identified via postoperative computed tomography in one of our patients is shown in Figure 3. 

When comparing the inner ascending aortic graft length, patients without collapse showed significantly shorter lengths than patients with AMDS collapse (30.0 ± 5.9 vs. 39.6 ± 10.9 mm; *p* = 0.029). However, the proximal and distal stent portions remained inflated and did not seem to be affected by the central collapse. The increased stretching of the stent portion may result in a decrease in AMDS diameter favoring stent collapse as demonstrated by the authors. In standard hemiarch replacement, the proximal part of the aortic arch is resected transversely toward the inner arch’s curvature, and the ascending aortic graft is shortened and resected with a corresponding shorter portion at the inner arch curvature. This approach aims to avoid the potential kinking of an elongated ascending graft. However, it may increase the tension of the AMDS toward the sinotubular junction, leading to a higher risk of stent collapse. Another potential risk factor may be the configuration of a gothic aortic arch or prominent aortic kinking of the arch and/or the descending aorta. Though no official contraindications exist for this scenario, AMDS implantation might be reconsidered under these circumstances to avoid stent collapse. Finally, aortic reshaping after the establishment of a pulsatile blood flow may lead to slight aortic elongation favoring stent collapse. On this basis, the authors proposed three key factors that might increase the risk for central stent collapse (Table 3). In our series of 100 AMDS implantations, we recognized the same phenomenon in three patients [6]. One of them was most likely caused due to the high tortuosity of the proximal descending aorta, which confirms the points mentioned by Luehr et al. [30]. One of three patients underwent successful endovascular dilatation of the collapsed stent. In the other two patients, endovascular dilatation was not a feasible option, but due to the uncovered stent design, no complications were observed in the further course and in-stent thrombosis was not evident during follow-up for these patients. However, we do recommend that in the case of AMDS collapse, medical anticoagulation therapy may be considered to avoid in-stent thrombosis [6,30].

## 6. Aortic Re-Intervention after AMDS Implantation

Early re-intervention due to malperfusion after surgery for DeBakey type I dissection is not uncommon and was also observed after AMDS implantation [6,15]. Most of these cases caused by branch vessel related malperfusion can successfully be treated using an endovascular approach. Current evidence is limited to a minimum regarding the further treatment of the downstream aorta in case of DeBakey type I dissection using thoracic endovascular aortic repair (TEVAR) after AMDS implantation [32]. While the FET serves as an excellent landing zone for the treatment of residual aortic dissection, the suitability of the AMDS for this concept is unknown [33,34]. In a case series with three patients, El-Andari et al. demonstrated excellent results for TEVAR following AMDS implantation [35]. The time from initial AMDS implantation to TEVAR ranged from three months to two years. Two patients presented with a progression of thoracic distal aortic aneurysm and one patient with a patent entry tear in the distal aortic arch causing dissection expansion in the absence of DANEs. All patients underwent TEVAR, and one required additional carotid-subclavian bypass. However, more information is needed for the treatment of chronic residual dissection using TEVAR after AMDS implantation.

No data in terms of aortic redo surgery after AMDS implantation are currently available. In our opinion, aortic redo surgery after AMDS implantation can be crucial and should only be performed in specialized aortic centers with corresponding expertise in endovascular and aortic arch surgery. Indications may be the dissection progression of the aortic arch, anastomotic leakage or aortic graft infection, including AMDS cuff infection, which might be a vulnerable spot in the case of graft infection. According to our experience, complete AMDS removal may only be feasible in the case of early redo surgery after AMDS implantation. During the later course, the stent portion is stuck in the aortic arch and descending aorta, carrying the risk of aortic damage if manual extraction is forced. In this case, the only possibility may be FET implantation into the AMDS combined with a debranching of the supra-aortic vessels. Due to the radial force of the AMDS, cutting the stent portion should not be performed to avoid an uncontrolled expansion of the stent. Though this might be of interest for extended arch surgery and the graft replacement of the aortic arch, we do not recommend cutting the stent. If urgently needed, a possible strategy could be applying several polypropylene sutures through the aortic wall to stabilize the stent frame and adapt it to the aortic wall tightly. This may avoid the uncontrolled expansion of the stent frame and allow for cuff removal and graft replacement beyond zone 0. An example is shown in Figure 4, where we performed cuff removal in one of our patients. No data are currently available regarding the incidence or impact of arch entries after AMDS implantation or on the risk of stent-induced new entries. These topics may be of upmost importance for the surgical community to identify patients at risk. Corresponding data are urgently needed.

## 7. Conclusions

Though experience and numbers are currently limited, the AMDS provides a promising and useful upgrade for standard hemiarch repair in the treatment of DeBakey type I dissection. Its use may be associated with a reduced risk for DANEs, positive aortic remodeling and the effective treatment of malperfusion. Preliminary studies show satisfactory short- and mid-term outcomes, considering that long-term data are urgently needed. Additionally, more data about standard hemiarch replacement compared to additional AMDS implantation are needed. Compared to total arch replacement using a FET, the AMDS may represent a valid alternative in the setting of acute DeBakey type I dissection—if no contraindications are present. This highlights the importance of careful preoperative planning including multiplanar computed tomography reconstructions to offer adequate aortic repair. Further information in terms of device failure and re-intervention after AMDS implantation is highly needed to identify patients at risk.

## Figures and Tables

**Figure 1 jcm-13-02593-f001:**
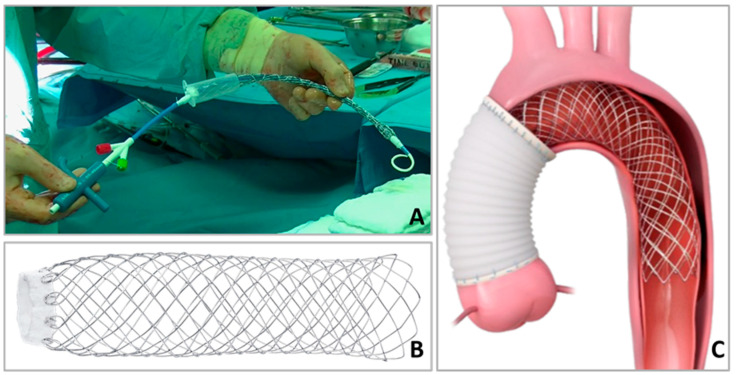
Graphical example of the AMDS device (used with the permission of Artivion^®^, Atlanta, GA, USA). (**A**) AMDS, including the delivery system before implantation. (**B**) Fully unfolded AMDS. (**C**) Hemiarch replacement and AMDS implantation at zone 0.

**Figure 2 jcm-13-02593-f002:**
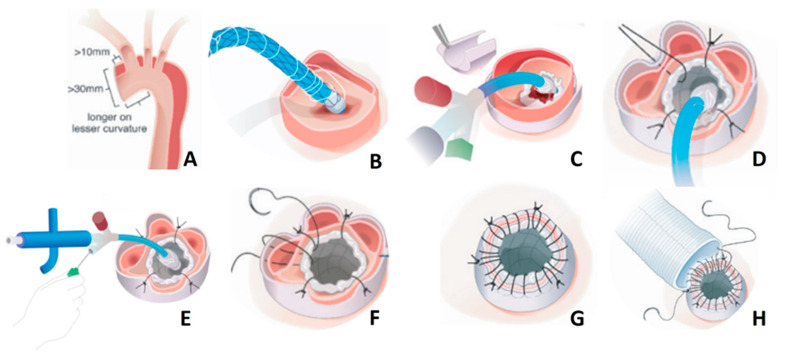
Schematic of the step-by-step implantation of the AMDS (used with the permission of Artivion®, Atlanta, GA, USA). In summary, the steps are as follows: (**A**) Leave > 10 mm distance to the innominate artery and longer on the lesser curvature when transecting the ascending aorta (the diameter at the distal anastomosis should be >30 mm). Store the AMDS in saline solution until implantation; (**B**) Insert the AMDS into the true lumen until the cuff reaches the plane of the transected aorta (if a guidewire is used, remove it before stent expansion); (**C**) Remove the plastic sheath to expose the polytetrafluorethylene felt; (**D**) Place an external polytetrafluorethylene felt strip around the aorta, and then place four single non-interrupting polypropylene sutures beginning at 6 o’clock, followed by 12, 3 and 9 o’clock to stabilize the cuff. This sandwich technique is highly recommended to avoid tearing the dissected aortic tissue; (**E**) While stabilizing the felt, unscrew the green cap counterclockwise to remove it completely. Pull back the sutures to expand the stent portion; (**F**) The delivery system can be removed once the stent is fully expanded and the tip of the delivery system is free from the stent. If tension appears, a stiff guidewire may be used to straighten the tip of the delivery system; (**G**) Perform a running suture in terms of a sandwich technique, consisting of the inner stent cuff, the aortic tissue and the outer felt. Avoid the enfolding of the inner cuff; (**H**) Finish the distal anastomosis by performing a running suture between the dacron graft and the sandwich cuff while ensuring to take all layers of the “felt–aortic–felt” complex with every stitch for the maximum seal of the distal anastomosis.

**Figure 3 jcm-13-02593-f003:**
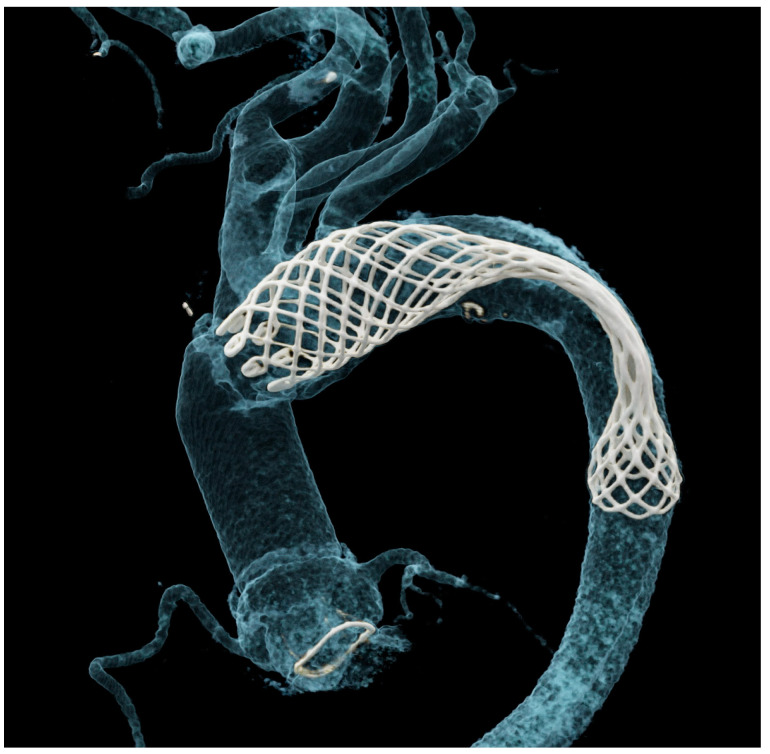
Postoperative computed tomography-based 3D reconstruction showing central AMDS collapse.

**Figure 4 jcm-13-02593-f004:**
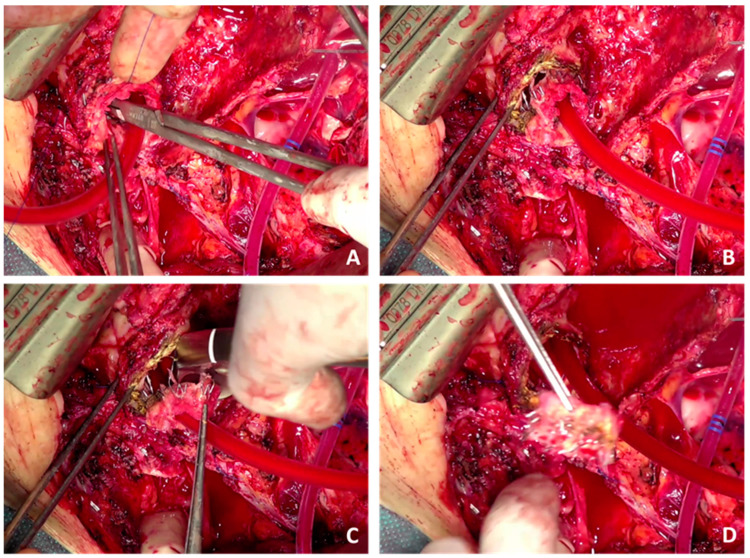
Example of AMDS shortening, including cuff removal. (**A**) Application of multiple single non-interrupting polypropylene sutures to stabilize the stent frame. (**B**) Careful dissection of the felt cuff. (**C**) Cutting the stent frame. (**D**) Complete removal of the cuff without an uncontrolled expansion of the stent.

**Table 1 jcm-13-02593-t001:** Indications and contraindications for AMDS implantation.

Indications	Contraindications
DeBakey Type I dissection	Aneurysm of the aortic arch or descending aorta
Primary entry in the ascending aorta or the aortic root	Entries in the aortic arch or descending aorta including supra-aortic vessels
	Connective tissue disorder (e.g., Marfan syndrome)
	Nickel (nitinol) allergy

**Table 2 jcm-13-02593-t002:** Current study results investigating outcomes after AMDS implantation in DeBakey type I dissection.

Author and Year	Number of Patients	Preoperative Malperfusion, n (%)	Thirty-Day Mortality, n (%)	(New *) Postoperative Stroke, n (%)	Device Failure, n (%)	DANE, n (%)	False lumen Thrombosis (Complete or Partial), n (%)
Bozso, 2022 [27]	n = 47	26 (56.5%)	6 (13%)	1 (4.8%) *	0 (0%)	0 (0%)	Zone 0: 19 (91%) Zone 1: 12 (60%) Zone 2: 15 (68%) Zone 3: 16 (68%) Zone 5: 16 (89%)
Montagner, 2022 [6]	n = 100	46 (46%)	18 (18%)	8 (8%) *	3 (3%)	n.a.	Zone 4: 67 (76%)
Luehr, 2022 [30]	n = 57	41 (72%)	9 (16%)	2 (4%) *	5 (8%)	n.a.	n.a.
White, 2023 [12]	n = 37	13 (35%)	5 (14%)	8 (21.6%)	n.a.	4 (12%)	Zone A: 24 (73%) Zone B1: 24 (73%) Zone B2: 26 (81%) Zone B3: 25 (81%) Zone C: 21 (68%)

Zone A = aortic arch; Zone B1 = proximal descending; Zone B2: mid-descending; Zone B3: distal descending; Zone C: infradiaphragmatic. “*“ defines “new“ postoperative stroke. This is why it is stated in brackets and is separated from the general postoperative stroke rate without “*“. This difference is often made in the aortic community.

**Table 3 jcm-13-02593-t003:** Potential risk factors for central AMDS collapse according to Luehr et al. [30].

Potential Risk Factors for Central AMDS Collapse		
Unfavorable anatomy	Device application	Aortic elongation
• Gothic aortic arch	• AMDS oversizing	• During reperfusion
• Aortic kinking	• Suboptimal aortic transection	
	• Increased proximal tension	

## Data Availability

The original contributions presented in the study are included in the article. Further inquiries can be directed to the corresponding author.

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
