# Peer review of "The Ascyrus Medical Dissection Stent: A One-Fits-All Strategy for the Treatment of Acute Type A Aortic Dissection?"

_jcm, 2024, doi:10.3390/jcm13092593_

Round 1

Reviewer 1 Report

Comments and Suggestions for Authors

I would like to thank authors for summarizing existing evidence of AMDS use in case of acute aortic dissection. However, I have comments on this:

Authors mentioned about Djumbodis system that could be used for the surgery for the acute aortic surgery. At glance Djumbodis system and AMDS have similarity. Therefore, I feel that more extensive comparative data with features of Djumbodis system and AMDS in terms of benefit of the latter should be presented in the manuscript.

Is there any room for AMDS implantation for chronic aortic dissection limited to aortic arch?

Author Response

Dear Reviewer 1,

thank you for your comments. We aimed to address all your points adequately. The comments, our answers and our changes can be found below.

Comment 1: Authors mentioned about Djumbodis system that could be used for the surgery for the acute aortic surgery. At glance Djumbodis system and AMDS have similarity. Therefore, I feel that more extensive comparative data with features of Djumbodis system and AMDS in terms of benefit of the latter should be presented in the manuscript.

Answer 1: Thank you for your comment. Indeed, we mentioned the Djumbodis Dissection Stent but did not point out clear differences. We added an additional section to the review.

Changes 1: Main differences between the Djumbodis Dissection System and AMDS may be the non-availability of the proximal cuff to avoid the formation of DANE and stent migration as well as the non-self-expanding stent capability which favors false lumen patency. Besides low numbers of implantations, available data on the long-term outcome of patients who received additional implantation of the Djumbodis Dissection Stent is limited to a minimum. Vendramin et al. summarized the long-term follow up and late complications of patients treated with the Djumbodis Dissection Stent and discovered high rates of late complications associated to the device, among them stent fracture and stent migration (18). The authors not only determined the insufficiency of this device but also advised that patients with a Djumbodis Dissection Stent undergo regular monitoring to mitigate the risk of potential catastrophic incidents stemming from device failure.

Comment 2: Is there any room for AMDS implantation for chronic aortic dissection limited to aortic arch?

Answer 2: Thank you for your comment. Currently there are no data available about using AMDS in the setting of chronic aortic dissection and we do not use it for this indication. We added a clarifying sentence to the manuscript.

Changes 2: In case of chronic aortic dissection, no experience is currently available in terms of AMDS implantation and therefore no reliable recommendations can be given so far. According to our experience, AMDS implantation should not be considered for the treatment of chronic aortic dissection, e.g. of the aortic arch.

Reviewer 2 Report

Comments and Suggestions for Authors

Article is well written and interesting although it doesn't bring a significant contribution to the field. 

Authors make a review about the Ascyrus Medical Dissection Stent

The work is interesting considering the novelty of the technique, but articles described are few and no personal data of authors are described. Not many significant reviews are available

Author could include some of their patients treated with ascyrus stent to make it more interesting

Conclusions are congruous

References are appropriate

Figure and table are satisfactory

Other comments: 

- define the abbreviations and continue to use them always

- hypothesize what should be the continuation of studies in this area

Author Response

Dear Reviewer 2,

thank you for your comments. We aimed to address all your points adequately. At this point of time, no review summarizing the current evidence and data about the AMDS is available. Some information may sound trivial, but we discovered that there is relevant lack of information and confusion in some parts of the cardiovascular surgical community about the usage of this novel device. Therefore, we aimed to provide an overview about the existing data, close knowledge gaps and clear up misunderstandings. In this context, we highlighted our own experiences and data as you recommended. Furthermore, we revised our abbreviations and used them consistently. Finally, we added some information what should be the continuation of studies in this area. 

Our changes can be found below. Changes are marked red in the manuscript.

Changes 1: A corresponding video demonstrating the step-by-step implantation and providing surgical tips and tricks was recently published by our group (10). 

Changes 2: Main differences between the Djumbodis Dissection System and AMDS may be the non-availability of the proximal cuff to avoid the formation of DANE and stent migration as well as the non-self-expanding stent capability which favors false lumen patency. Besides low numbers of implantations, available data on the long-term outcome of patients who received additional implantation of the Djumbodis Dissection Stent is limited to a minimum. Vendramin et al. summarized the long-term follow up and late complications of patients treated with the Djumbodis Dissection Stent and discovered high rates of late complications associated to the device, among them stent fracture and stent migration (18). The authors not only determined the insufficiency of this device but also advised that patients with a Djumbodis Dissection Stent undergo regular monitoring to mitigate the risk of potential catastrophic incidents stemming from device failure.

Changes 3: According to our expertise, not only entries in the aortic arch or descending aorta but also in the supra-aortic vessels may contribute to a perfused false lumen after AMDS implantation, leading to aortic growth and a high risk for complex redo surgery.

Changes 4: In case of chronic aortic dissection, no experience is currently available in terms of AMDS implantation and therefore no reliable recommendations can be given so far. According to our experience, AMDS implantation should not be considered for the treatment of chronic aortic dissection, e.g. of the aortic arch.

Changes 5: Current evidence is limited, but in a series of 16 patients, we were able to demonstrate satisfactory results in terms of supra-aortic vessel restoration and reached 100% regression of totally occluded supra-aortic branches after the AMDS was implanted (26). Later on, we published the currently largest available series of AMDS implantations for treatment of DeBakey type I dissection including 100 patients (8). 

Changes 6: An example of AMDS collapse identified by postoperative computed tomography in one of our patients is shown in figure 3. 

Changes 7: However, we do recommend that in case of AMDS collapse, medical anticoagulation therapy may be considered to avoid in-stent thrombosis (8, 30).#

Changes 8: According to our experience, complete AMDS removal may only be feasible in case of early redo surgery after AMDS implantation. 

Changes 9: In this case, the only possibility may be FET implantation into the AMDS combined with debranching of the supra-aortic vessels. 

Changs 10: Though this might be of interest for extended arch surgery and graft replacement of the aortic arch, we do not recommend cutting the stent.

Changes 11: An example is shown in figure 4, where we performed cuff removal in one of our patients. No data are currently available regarding the incidence or impact of arch entries after AMDS implantation as well as the risk of stent induced new entries. These topics may be of upmost importance for the surgical community to identify patients at risk. Corresponding data are urgently needed.